# Identification of Cold Tolerance Transcriptional Regulatory Genes in Seedlings of *Medicago sativa* L. and *Medicago falcata* L.

**DOI:** 10.3390/ijms251910345

**Published:** 2024-09-26

**Authors:** Qi Wang, Jianzhong Wu, Guili Di, Qian Zhao, Chao Gao, Dongmei Zhang, Jianli Wang, Zhongbao Shen, Weibo Han

**Affiliations:** 1Institute of Forage and Grassland Sciences, Heilongjiang Academy of Agricultural Sciences, Harbin 150086, China; m18846603788@163.com (Q.W.); wujianzhong@haas.cn (J.W.); gaochaopdf2000@163.com (C.G.); zhd_mei@163.com (D.Z.); wangjianlivip@126.com (J.W.); shzhb1973@126.com (Z.S.); 2College of Life Science and Technology, Harbin Normal University, Harbin 150025, China; 3Institute of Industrial Crops, Heilongjiang Academy of Agricultural Sciences, Harbin 150086, China; diguili59@163.com; 4Cultivation and Farming Research Institute, Heilongjiang Academy of Agricultural Sciences, Harbin 150086, China; zhaoqian0401@sina.com

**Keywords:** DEGs, cold tolerance, transcriptome, *Medicago sativa* L., *Medicago falcata* L.

## Abstract

Alfalfa species *Medicago sativa* L. (MS) and *Medicago falcata* L. (MF), globally prominent perennial leguminous forages, hold substantial economic value. However, our comprehension of the molecular mechanisms governing their resistance to cold stress remains limited. To address this knowledge gap, we scrutinized and compared MS and MF cold-stress responses at the molecular level following 24 h and 120 h low-temperature exposure (4 °C). Our study revealed that MF had superior physiological resilience to cold stress compared with MS, and its morphology was healthier under cold stress, and its malondialdehyde content and superoxide dismutase activity increased, first, and then decreased, while the soluble sugar content continued to accumulate. Transcriptome analysis showed that after 120 h of exposure, there were different gene-expression patterns between MS and MF, including 1274 and 2983 genes that were continuously up-regulated, respectively, and a total of 923 genes were included, including star cold-resistant genes such as *ICE1* and *SIP1*. Gene ontology (GO) enrichment and Kyoto Encyclopedia of Genes and Genomes (KEGG) pathway analyses revealed numerous inter-species differences in sustained cold-stress responses. Notably, MS-exclusive genes included a single transcription factor (TF) gene and several genes associated with a single DNA repair-related pathway, whereas MF-exclusive genes comprised nine TF genes and genes associated with 14 pathways. Both species exhibited high-level expression of genes encoding TFs belonging to AP2-EREBP, ARR-B, and bHLH TF families, indicating their potential roles in sustaining cold resistance in alfalfa-related species. These findings provide insights into the molecular mechanisms governing cold-stress responses in MS and MF, which could inform breeding programs aimed at enhancing cold-stress resistance in alfalfa cultivars.

## 1. Introduction

*Medicago sativa* L. (MS), commonly known as alfalfa, is a perennial herbaceous plant belonging to the pea family (Fabaceae) within the order Fabales (legumes). Alfalfa, renowned for its exceptional nutrient content, is the most extensively cultivated legume forage globally. However, its wild relative, *Medicago falcata* L. (MF), exhibits superior resistance to drought, cold, and infertile soil conditions [1], making MF a valuable parent material in breeding programs aimed at enhancing MS resilience to harsh growing environments [2]. In China, alfalfa cultivation primarily occurs in North China, where recurring frost damage of seedlings or plants following low-temperature exposure presents significant challenges to alfalfa production and dissemination, impacting animal husbandry [3]. This vulnerability results in substantial economic losses, necessitating a deeper understanding of the impact of low-temperature stress on alfalfa and the molecular mechanisms governing its response to cold stress.

Cold stress induces various overt detrimental effects, including surface lesions, discoloration, tissue destruction, and accelerated aging [4,5], while at the cellular level, it disrupts the cell membrane, leading to electrolyte leakage and malondialdehyde (MDA) production [6], as well as to changes in gene transcription and protein synthesis [7]. Ultimately, these changes hinder plant growth, with the degree of impact varying depending on a plant’s genetic composition and stress tolerance.

Current studies have shown that alfalfa can scavenge toxic free radicals and reactive oxygen species (ROS) by activating the antioxidant defense system in response to cold stress [8], or alleviate oxidative stress by enhancing the level of autophagy, such as Zhao et al. [9] found that *MsATG13* overexpression can increase the level of autophagy, increase the content of proline and antioxidant enzymes, and enhance the antioxidant defense capacity under cold-stress conditions. Meanwhile, alfalfa utilizes osmotic regulatory substances to enhance cell fluid content and reduce cell osmotic potential, consequently lowering the cellular freezing point; additionally, they bolster the protoplasm’s capability to alleviate cell damage [10,11]. In addition, plant hormones emerge as crucial modulators in the regulation of cold-stress signal transduction. For instance, abscisic acid (ABA) controls the expression of COR genes by modulating C-repeat binding factor (CBF) transcription [12], and it has been speculated that C-repeat binding factor-like (CBFl) may play the same role in alfalfa [13]. Furthermore, several differentially expressed genes (DEGs) potentially related to autumn dormancy have been identified using the synthetic backcrossing strategy (SBS). This strategy utilizes crown buds and whole alfalfa plants as materials, providing valuable resources for future functional genomics studies related to freezing stress [14,15]. The latest study found that chloroplasts store nutrients through photosynthesis to resist cold stress, and some genes are down-regulated or edited less efficiently [16].

However, most such studies have focused on a single species of alfalfa exposed to stress for less than 5 days, limiting the exploration of unique or shared cold-resistance genes and molecular mechanisms across species. In order to address the above knowledge gaps, cold stress was applied to MS and MF for 24 h and 120 h, respectively, at 4 °C. The physiological and metabolic changes of MS and MF in response to cold stress were measured, and then Gene Ontogeny (GO) enrichment and Kyoto Encyclopedia of Genes and Genomes (KEGG) pathway analysis were performed based on pan-transcriptional sequencing results. The purpose of this study was to independently analyze the genetic basis of the cold-stress response of MS and MF at the transcriptional level, and to compare them based on the results, aiming to provide potential genetic resources and valuable references for enhancing the cold resistance of alfalfa.

## 2. Results

### 2.1. Effects of Cold Stress on Morphological and Physiological Responses of MA and MF Seedlings

In the control group, the seedlings of both MS and MF plants had healthy leaves, and the plants of MF were shorter and had smaller leaves compared to MS. After 24 h of cold treatment, a small number of MS leaves showed signs of wilting and tissue softening (in the red circle of Figure 1A), while most of the MS plants in the 120 h treatment group showed slight stem drooping and a large number of leaves drooping, wilting, and yellowing, and only a few remained healthy (in the blue circle of Figure 1A), which was less common in MF leaves. With the increase of the duration of cold stress, leaf wilting and growth inhibition intensified, and the morphological changes of seedlings at the MS three-leaf stage were greater than those of MF.

MDA levels significantly increased in both the MS and MF groups after 24 and 120 h of cold stress (*p* < 0.05), surpassing those of the control group (Figure 2A). SOD activity showed a significant increase after 24 h of cold stress (*p* < 0.05), followed by a decrease after 120 h (*p* < 0.05), though it remained higher than in the control group (Figure 2B). The SS content continued to rise in both alfalfa species under cold stress, although the SS content of the MS group was not significantly higher than that of the control group after 24 h (*p* > 0.05) (Figure 2C). Notably, MF exhibited a lower MDA content and higher SOD and SS contents than MS under cold-stress conditions for 0, 24, and 120 h.

### 2.2. RNA Sequencing and Transcriptome Assembly, Comparison, and Analysis

Eighteen cDNA libraries were constructed, yielding a total of 109.14 Gb of raw reads with an average of 6.06 Gb of reads per sample (Table 1). After filtering to remove redundant reads, a total of 108.40 Gb of clean reads were obtained, averaging 6.02 Gb per sample, yielding a mean Q30 of 93.96% and a mean Q20 of 98.02%. Alignment with the alfalfa reference genome resulted in an estimated 32,595,076 to 41,194,666 successfully mapped clean reads, corresponding to an overall comparison rate ranging from 75.53% to 79.92%. Notably, the comparison rate of clean reads with unique genomic locations ranged from 71.20% to 76.20%, while those with multiple locations ranged from 3.58% to 4.41%. These findings highlight the high fidelity of the transcriptome data, enabling subsequent analyses.

Statistical analysis of the read distribution relative to the reference genome (Appendix A) revealed that transcriptome data were predominantly mapped to exonic regions.

Both alfalfa species exhibited significant responses to cold stress, with a total of 57,080 DEGs detected, including 7915 encoded by novel genes (Appendix A). The results of the comprehensive statistical analysis of gene-expression levels demonstrated consistent and uniformly distributed gene-expression levels across biological replicates, affirming the high quality of the 18 sequenced samples (Table 2 and Appendix A).

The results of principal component analysis (PCA) showed that the biological replicates were almost identical between the treatment groups, and in subsequent analyses, we will focus on individuals with good reproducibility as references (Figure 3). The correlation between all cold-stress groups was strong, and the Pearson correlation coefficient was consistently above 0.9 (Appendix A). It is worth noting that the correlation between the control group and the cold-stress group was always lower than 0.9, indicating that there were significant differences in gene-expression patterns between groups under cold stress.

The DEGs screening results (Figure 4A) highlight distinct responses to cold stress between the MS and MF cold-stressed groups at 24 h and 120 h compared to the control group. The comparison between the MS and control groups revealed 3175 DEGs (1176 up-regulated 1999 down-regulated) and 3070 DEGs (1508 up-regulated and 1562 down-regulated) at 24 h and 120 h, respectively (Appendix A). Similarly, the comparison between the MF and control groups yielded a higher number of down-regulated genes than up-regulated genes, including 5809 DEGs (2714 up-regulated and 3095 down-regulated) and 4962 DEGs (2403 up-regulated and 2559 down-regulated) at 24 h and 120 h, respectively (Appendix A).

Remarkably, inter-group comparisons revealed a higher prevalence of down-regulated genes across groups. Furthermore, 1274 genes in MS and 2983 genes in MF exhibited sustained significant regulation throughout the 120 h cold-stress period, denoted as MS-p and MF-p, respectively (Figure 4B, Appendix A).

Differences in gene expression between MS and MF were evident, with 923 co-expressed genes identified as common cold-stress response genes across both species (Appendix A). Notably, MF demonstrated a greater number of cold-stress response genes, with 2060 genes continuously regulated compared to 351 in MS. However, both varieties exhibited overlapping responses to cold stress, as indicated by the numerous cold-stress response genes common to both species.

Further analysis revealed that throughout the 120 h cold stress period, two genes were up-regulated in MS and down-regulated in MF, while one gene was down-regulated in MS and up-regulated in MF (Figure 4C). These findings suggest that nuanced regulatory mechanisms underlie the alfalfa response to cold stress.

### 2.3. Enrichment Analysis of DEGs

After annotation against the GO database, DEGs consistently implicated in the 120 h cold stress tolerance process in both MS and MF were categorized into three distinct classes (Figure 5, Appendix A). The most abundant GO terms for MS and MF in biological processes were “oxidation-reduction process (GO:0055114)” and “external encapsulating structure organization (GO:0045229)”, respectively. And the most numerous GO items in their cellular components and molecular functions were “membrane (GO:0016020)” and “catalytic activity (GO:0003824)”. Among the top 20 GO terms in functional categories, Biological Process, Cellular Component, and Molecular Function, MS and MF shared 1, 14, and 5 common GO terms, respectively.

Moreover, the 1274 genes consistently involved in 120 h cold-stress resilience in MS and MF were mapped to 30 and 44 KEGG pathways, respectively (Figure 6, Appendix A). The three pathways with the highest number of genes in MS and MF are “Metabolic pathways”, “Protein processing in endoplasmic reticulum”, and “Plant hormone signal transduction”. Notably, all identified MS pathways except for “Base excision repair” were observed in MF, with 29 pathways shared by MS and MF.

### 2.4. TF Family Analysis

Functional annotation revealed the presence of 26 and 35 TF families among all significantly expressed DEGs implicated in the 120 h cold-stress responses in MS and MF, respectively (Appendix A). Notably, 19 TF families were shared between both species.

In MS, the highest TF counts were associated with the AP2-EREBP and bHLH TF families, each boasting nine TFs. Following closely were the ARR-B and WRKY families, each comprising eight TFs. Conversely, in MF, the AP2-EREBP TF family predominated, with 25 TFs, followed by the ARR-B, bHLH, and NAC TF families, with 24, 14, and 13 TFs, respectively (Figure 7).

Remarkably, the AP2-EREBP family TFs comprised the most abundant shared TFs, with seven TFs overlapping between the two species. Additionally, the bHLH and WRKY families contributed six TFs each, further underscoring their significance in the cold-stress response across MS and MF.

### 2.5. RNA-seq Expression Validation by RT-qPCR

Gene-expression profiling was performed using real-time reverse transcription quantitative PCR (RT-qPCR) to quantitatively determine the reliability of our transcriptome data. For this evaluation, six candidate cold-stress DEGs were monitored. The expression profiles of these selected DEGs were determined using qRT-PCR of duplicate biological samples processed in parallel. The results were in agreement with the RNA-seq values obtained using each method differed by a log2-fold difference (Figure 8). RNA-seq and qRT-PCR displayed a positive correlation with a Pearson coefficient R^2^ = 0.9981 (*MsUBC2*) or R^2^ = 0.9982 (*MsTUB*). Our results indicated that qRT-PCR expression profiles for the six selected DEGs were generally consistent with RNA-seq results.

## 3. Discussion

### 3.1. Distinct MS and MF Morphological and Physiological Changes in Response to Low-Temperature Stress

Under normal growth conditions, MF plants typically exhibit a shorter height, smaller overall size, and smaller leaves than MS, with a creeping stem lying close to the ground as opposed to the upright stems of MS. Following stress exposure, MF leaves appeared heathier than those of MS, suggesting that MS may be more susceptible to cold stress compared to MF. This observation aligns with that of previous studies indicating that alfalfa varieties characterized by a lower plant height, shorter leaf length, and reduced biomass tend to demonstrate enhanced cold-stress resistance [17,18,19].

Cold hardiness in plants involves intricate physiological mechanisms aimed at mitigating damage, including alterations in plasma membrane properties, up-regulation of ROS-scavenging enzymes, and augmentation of osmoregulatory mechanisms [20,21]. In this study, we investigated the physiological and metabolic changes in two alfalfa species before and after exposure to cold stress.

Comparisons between healthy MS and MF plants revealed lower MDA levels and higher SOD and SS levels in MF plants compared to MS plants under normal conditions. However, both species exhibited increases in MDA levels following 24 h and 120 h of exposure to 4 °C stress, indicating membrane damage. Remarkably, MF consistently maintained lower MDA levels than MS, suggesting superior antioxidant capacity in this species. Meanwhile, SOD levels initially rose then declined in both species, indicating activation of the defense system in response to cold stress, although the protective efficacy of defense enzymes appeared limited, consistent with previous reports [22]. In contrast, the SS content of both species showed an upward trend, and that of MF showed a more significant increase under cold stress, indicating that soluble sugars accumulated under cold stress, alleviating water deficit and enhancing cold resistance under cold stress, which is consistent with previous studies [23].

Furthermore, MF consistently exhibited higher levels of defense enzymes and osmoregulatory substances compared to MS following 24 h and 120 h of 4 °C exposure, while maintaining lower MDA levels. These observations underscore the superior cold resistance of MF compared to MS, corroborating findings from previous studies [24,25].

### 3.2. Transcriptome Analysis of MS Seedlings under Low-Temperature Stress

Alignment of reads to various reference genome features, including introns and intergenic regions, offered insights into dynamic gene-expression changes in MS and MF under low-temperature stress. The overall comparison rates between the materials selected in this study and the alfalfa reference genome ranged from 75.53% to 79.92%, likely due to the cold-region specificity of the plant material we used being produced at 45 degrees north latitude. In this study, we used the group with better reproducibility of PCA results as the main analysis object, and the other group as a reference. Next, we will further refine the mechanism of alfalfa resistance to cold at the transcriptional level and increase the biological and technical replicates to improve the quality of the data. Intron region alignments may have stemmed from residual pre-mRNA or intron retention during variable shearing, while alignments with intergenic regions may have resulted from the transcription of unknown genes or non-coding RNA, providing a basis for subsequent expression analyses.

In MS, 1274 genes exhibited continuous regulation under 120 h of cold stress. GO enrichment analyses of these genes revealed significant enrichment of antioxidant-related terms such as “oxidation-reduction process” and “peroxisome”, indicating the key role of antioxidants in MS resistance to cold stress, consistent with the observed elevated SOD values after low-temperature exposure. Additionally, GO terms like “protein phosphorylation”, “photosynthesis”, and “membrane” were significantly enriched, aligning with prior transcriptome sequencing analyses of alfalfa’s response to cold stress [26]. Among them, protein phosphorylation is able to directly cause Ca^2+^ signaling coupling to the cold-specific transduction pathway [27]; chloroplasts are the site of photosynthesis in plants, but low temperature will destroy the chloroplast structure, reduce the chlorophyll content, and, thus, reduce the photosynthetic efficiency, which is also an important cause of plant yellowing [28]; the metabolic level of plants will change when they are subjected to cold stress, and the up-regulation of the expression of catalytic activity-related genes will solve this problem to a certain extent [29]; and membranes are the first to sense low-temperature stress and, in response to [30], plants resist cold stress by maintaining membrane homeostasis by maintaining the unsaturation of membranous substances during cold stress [31].

KEGG pathway analysis highlighted the significant involvement of the “Peroxisome” pathway in MS’s cold-stress response, affirming the pivotal role of defense enzymes in MS cold-stress tolerance. Moreover, the significant enrichment of “Plant hormone signal transduction” underscores the central regulatory role of hormones in MS cold-stress responses [14], while other enriched pathways identified in this study corroborate previously reported findings, including “Isoflavonoid biosynthesis”, “Circadian rhythm—plant”, and “Glycerophospholipid metabolism”. Among them, isoflavones have antioxidant properties and can scavenge ROS under cold stress [32]; CBF expression is regulated by the circadian clock through the action of a central oscillator and also by day length (photoperiod) [33]; and glycerophospholipids are the main components of the plasma membrane, which help to maintain plasma membrane fluidity and cell function under low-temperature stress [34].

Notably, in this study, a total of 78 transcription factors from 26 families were found in significantly expressed DEGs, including *RAP2-1*, *ABF2*, *TIFY6B*, *GATA8*, and *SPL16*, encoding TFs belonging to AP2-EREBP, bZIP, Tify, C2C2-GATA, and SBP TF families that exhibited high-level expression. *RAP2-1* encodes the ethylene-responsive TF *RAP2-1*, a downstream regulator of C-repeat binding factors (CBFs) [35] that has been validated as a classical cold-responsive gene transcript in soybean [36]. *ABF2* encodes bZIP TF 46, which acts as an ABA-responsive element with a crucial role in cold resistance of crops, including strawberries [37]. *TIFY6B*, a member of the plant-specific TIFY TF family of adversity response regulators, encodes *TIFY6B*. In chili peppers, *TIFY6B* acts as a cold-stress-responsive element with shared functions with TFs CaTIFY7 and CaTIFY10b, members of the same TF family, which have been reported to enhance plant cold tolerance by inducing the expression of cold stress-related genes and triggering the ROS response [38]. *GATA8*, encoding GATA TF 8, has also been implicated in the cold-stress responses of Arabidopsis and rice [39], further underscoring the multifaceted regulatory network involved in MS’s response to cold stress.

### 3.3. Transcriptome Analysis of MF Seedlings under Low-Temperature Stress

A total of 2983 genes displayed continuous regulation during 120 h of cold stress. GO enrichment analysis revealed the significant enrichment of antioxidant-related terms such as “oxidoreductase complex”, “peroxisome”, and “peroxisomal part”, highlighting the crucial role of antioxidants in MF cold-stress tolerance. Moreover, the enrichment of cell wall-related terms like “pectinesterase activity”, “cell wall modification”, and “cell wall” underscores the importance of the dynamic cell wall polysaccharide network in providing stability and protection to plants under cold-stress conditions. Pectin esterases, ubiquitous cell wall-associated enzymes, which play key roles in shaping cell wall mechanical properties and porosity through the demethylation of pectin [40], have also been implicated in stress responses [41].

Furthermore, MF genes were enriched for numerous KEGG pathway terms related to resistance to cold stress, including “Peroxisome”, which involves defense enzymes, and “Plant hormone signal transduction”, which involves genes associated with the synthesis of JA, ABA, gibberellins, oleoresin steroids, and cytokinins, thereby affirming the role of phytohormones in the abiotic-stress response. Additionally, the pathway terms “Isoflavonoid biosynthesis” and “Circadian rhythm—plant” were also significantly enriched, consistent with previous studies.

In MF, a total of 154 TFs belonging to 35 TF families with potential roles in the cold stress response were identified, with genes encoding members of the plant-specific AP2-EREBP TF family (*DREB1A*, *ERF110*, *ABR1*, and *WRI1*) and the ARR-B family (*APRR1*) exhibiting notably high expression levels. The AP2-EREBP family is known for its roles in plant development and responses to diverse biological and environmental stresses [42]. For example, overexpression of *DREB1A*, which encodes DREB-like protein 1A (*DREB1A*), endows plants with an enhanced tolerance to salinity, cold, and drought stress [43]. Similarly, *ERF110*, which encodes the ethylene-responsive TF *ERF110*, is involved in the rye drought stress response [44]. Another notable gene, *ABR1*, encodes the ethylene-responsive TF *ABR1*, which acts as an ABA-responsive repressor in *Arabidopsis thaliana* that participates in responses to cold, high salt, and drought stress [45]. *WRI1* encodes a well-known TF that regulates crop seed oil biosynthesis [46] and the expression of nonspecific lipid transfer proteins in response to nematode infections [47]. Additionally, *APRR1* encodes the two-component response regulator *APRR1*, a well-known TF that participates in regulating circadian rhythms [48].

### 3.4. Comparison of the MS and MF Transcriptomes under Low-Temperature Stress

The ICE1-CBF-COR pathway is the most widely studied cold-stress signaling pathway, which activates the expression of CBF through INDUCER OF CBF EXPRESSION 1 (ICE1) binding to the promoter region of CBF gene under cold-stress conditions, thereby improving the frost resistance of plants, and the role of this pathway has been emphasized in the participation of wheat crops in abiotic-stress response [49]. In this study, we analyzed the differential genes of MS and MF, and we found that the cold-resistance process of the materials used in this experiment was related to ICE1-CBF-COR pathway. The MsG0380016999 gene encoding the transcription factor ICE1 isoform X1 was significantly up-regulated in both alfalfa species after 24 h of cold stress, and it remained up-regulated until 120 h of cold stress. In addition, MsG0480021424, encoding CBF5, was significantly up-regulated after MF was subjected to cold stress for 24 h. The results of this study are consistent with the results suggesting that the overexpression of the ICE1 and CBF genes can confer high cold tolerance in plants [50,51], indicating that the ICE1–CBF–COR pathway plays an important role in the process of cold-stress resistance in alfalfa, especially in MF.

Aquaporin is the main channel of water transport in plant cells, and abiotic stress dehydrates cells by changing the water potential; it has been shown that aquaporin actively participates in regulating plant stress resistance [52]. In this study, we identified four isoforms of aquaporin: plasma membrane intrinsic proteins (PIPs), tonoplast intrinsic proteins (TIPs), nodulin 26-like proteins (NIPs), and small basic intrinsic proteins (SIPs). MsG0180000871 and MsG0780041015, which regulate TIP1 and TIP4, were significantly down-regulated in MS and MF after cold stress, while SIP1, which is regulated by MsG0780040029 and MsG0680031712, continued to be significantly up-regulated in MF, which was consistent with the situation found in maize and rice seedlings [53,54]. In addition, we found that MsG0180000871 was significantly down-regulated in MF earlier than it was in MS, indicating that MF responded more quickly to cold stress than MS. We also found that MsG0580029158 and MsG0480020797, which regulate PIP1 and PIP2, respectively, were down-regulated after 120 h of cold stress in both alfalfa species. Western blot analysis confirmed that Arabidopsis PIP1; 4 and PIP2; 5 was specifically up-regulated during cold acclimation, thereby increasing protein mass and total pip [55]. Our results of this experiment are inconsistent, and we suspect that the plasma membranes of both alfalfa species may have been damaged due to prolonged low-temperature stress, thus affecting the gene function. Interestingly, differential expression of MsG0780036261 encoding NIP2 and MsG0380011693 was detected in both alfalfa species; the former has been down-regulated and the latter has been up-regulated, but MsG0780036261 has been down-regulated in MS and less so in MF. Studies have shown that CaNIP1-5 and CaNIP4-1 are highly expressed in the aerial tissues of chickpea after low-temperature stress [56]; Overexpression of the VvNIP gene in grapes improved cold resistance [57]. However, some studies have suggested that the down-regulated expression of CsNIP5 in tea plants under low-temperature stress may be related to the ABA-dependent osmotic stress response [58]. There has still been no research on alfalfa, and we prefer that NIP2 should be gradually up-regulated under cold stress, while the reason for the down-regulation of MsG0780036261 in this study remains to be investigated.

Statistical analysis revealed a significantly higher number of genes that were consistently significantly regulated under 120 h of cold stress in MF compared to MS, encompassing most identified MS cold-stress resistance genes. This finding suggests that MF exhibits greater cold resistance than MS.

While most pathways enriched in MS were also present in MF, the absence of the “Base excision repair” pathway in MF is noteworthy, given its crucial role in protecting genomes from damage caused by intrinsic and environmental factors [59]. Several pathways that are only significantly expressed in MF have been shown to be associated with cold stress, including “Steroid biosynthesis” [60], “Fatty acid degradation” [61], “Amino sugar and nucleotide sugar metabolism” [62], “Phenylpropanoid biosynthesis” [63], “Oxidative phosphorylation” [64], “DNA replication” [65], “Nucleocytoplasmic transport” [66], “alpha-Linolenic acid metabolism” [67], “Nucleotide excision repair” [59], “RNA polymerase” [68], “Betalain biosynthesis” [69], “Protein export” [70], “Basal transcription factors” [71], and “Endocytosis” [72], among others. Additionally, pathways such as “Stilbenoid, diarylheptanoid, and gingerol biosynthesis” have reported roles in drought and alkali stress responses [73,74]. The “Basal transcription factors” pathway encompasses TAF12B, which encodes a TF belonging to the plant-specific ARR-B TF family and is known for its reported pivotal roles in plant processes [75], including the regulation of plant root growth in *A. thaliana* [76]. These two pathways may also contribute to the cold-stress response in alfalfa. In addition, TFs belonging to the HB TF family, which have been shown to coordinate phytohormone production during biotic stress responses and mechanical injury in sunflower [77], responded consistently to cold stress only in MS but not in MF, prompting us to speculate that HB family member expression in MS may participate in abiotic-stress responses.

Intriguingly, BSD, CSD, E2F-DP, EIL, HSF, LIM, LOB, PLATZ, and Trihelix were exclusively represented in the DEGs of MF. Particularly noteworthy is the PLATZ family, consisting of plant-specific zinc-finger TFs pivotal for plant growth, development, and responses to abiotic stresses. Moreover, in alfalfa, members of the Trihelix TF family have been implicated in responses to salt, cold, and drought stress [78,79], whereas copper/zinc SOD (CSD) TFs have been shown to be induced by cold stress [80]. These discoveries collectively suggest the significant contributions of PLATZ, Trihelix, and CSD TF family members to the sustained MF response to cold stress. Regarding other TF families, research on BSD TFs suggests their involvement in plant growth and development, particularly in fruit ripening [81]. Similarly, adjacent E2F-DP TF sequences may contribute to glutathione transferase-related detoxification functions. Additionally, the EIL TF family plays a pivotal role in the ethylene signaling pathway, which governs various plant growth and developmental processes, and responds differentially to dehydration, salinity, and phosphate starvation in soybean [82], while Lin-11, Isl-1, and Mec-3 (LIM) domain TFs are crucial in regulating fundamental plant biological processes. Although these TF families have not been directly linked to cold-stress responses, our findings suggest that they contribute to MF abiotic-stress tolerance by sustaining the 120 h cold-stress response in this species. Heat shock TF (HSF) family members are vital in plant growth, development, and stress responses, as evidenced by their negative roles in maize salt and drought-stress responses [83]; in addition, their antagonistic effects on methyl jasmonate (MeJA)-induced cold tolerance in banana [84]. Therefore, HSF TFs likely play negative roles in various abiotic-stress responses that include a significant role in the MF response to cold stress.

In addition to the abovementioned genes, our study identified two novel genes encoding the MtN19-like and DUF1997 family proteins that displayed consistent up-regulation in response to cold stress. Additionally, the *GmSGT3* gene, encoding soyasaponin III rhamnosyltransferase, exhibited up-regulation in MF and down-regulation in MS under cold stress conditions. While the functions of the former two proteins remain unreported, *GmSGT3* expression has been linked to saponin accumulation in soybean [85]. Nevertheless, despite their differential regulation in response to cold stress in MS and MF, they hold potential as cold-stress-responsive genes.

Collectively, our findings highlight the nuanced differences between MF and MS in their sustained response to cold stress. While members of the AP2-EREBP, ARR-B, and bHLH TF families were highly expressed in both species, other TFs identified in MS and MF cold-stress transcriptomes also played crucial roles in regulating MF and MS transcriptional cold-stress responses. Therefore, we hypothesize that members of these three TF families and other TF families play pivotal roles in sustaining cold resistance across alfalfa species beyond those investigated in this study.

## 4. Materials and Methods

### 4.1. Plant Materials and Stress Treatment

This research was conducted at the Institute of Forage and Grassland Sciences, Heilongjiang Academy of Agricultural Sciences, located in Harbin, China (longitude: 126.53°, latitude: 45.80°). MS and MF experimental materials were grown under controlled conditions that included a photoperiod of 16 h of light followed by 8 h of darkness and a temperature of 22 °C. At the three-leaf stage of growth, alfalfa seedlings were subjected to cold stress in an artificial climate incubator set to maintain a light-dark cycle of 16-h light/8-h dark and a temperature of 4 °C. Three treatment groups were established for both MS and MF seedlings, consisting of two cold treatment durations of 24 h and 120 h, along with a control group at 0 h. For each treatment group, three biological replicates were carried out. Subsequently, all seedlings were harvested simultaneously then seedlings were photographed.

### 4.2. Physiological Index Determination

In this study, changes in Malondialdehyde (MDA), Superoxide dismutase (SOD), and Soluble sugar (SS) levels served as indicators for assessing both the extent of cell membrane damage and the ability of plants to adapt to cold stress. A total of 120 seedlings (20 seedlings per replicate for each species) were collected for the physiological experiments, with all treatment groups harvested at corresponding time points.

MDA content was determined using an MDA assay kit (Suzhou Keming Biotechnology Co., Ltd., Suzhou, China). MDA was condensed with thiobarbituric acid (TBA) to form a red product with a maximum absorption peak at 532 nm, and the content of lipid peroxide in the sample could be estimated after color comparison. At the same time, the absorbance at 600 nm was measured, and the difference between the absorbance at 532 nm and 600 nm was used to calculate the MDA content.

SOD content was determined using an SOD assay kit (Suzhou Keming Biotechnology Co., Ltd., Suzhou, China). Superoxide anion (O^2−^) is produced by xanthine and xanthine oxidase reaction system, and O^2−^ can reduce nitrogen blue tetrazolium to form blue formazan, which is absorbed at 560 nm. SOD can scavenge O^2−^, thereby inhibiting the formation of formazan; The darker the blue color of the reaction solution, the lower the SOD activity, and the higher the activity.

SS content was determined using an SOD assay kit (Suzhou Keming Biotechnology Co., Ltd., Suzhou, China). Based on the anthone method, furfural or hydroxymethylfurfural is further condensed with anthone reagent to produce a blue-green substance, which has the maximum absorption at the wavelength of 620 nm in the visible region, and its light absorption value is proportional to the content of sugar within a certain range.

Each experiment was repeated at least three times. Data analysis involved the use of Shapiro-Wilk test and one-way ANOVA test in SPSS v26, respectively, and plotting of data using Origin v2021.

### 4.3. RNA Extraction and cDNA Library Construction

After extracting total RNA from the samples, rRNA was removed, mRNA was enriched, and double-stranded cDNA was synthesized through reverse transcription. Subsequently, double-stranded ends were repaired and ligated then the resulting ligated DNA products were amplified via PCR to construct the cDNA library. To ensure sequencing quality, we assessed RNA integrity and DNA contamination of samples using agarose gel electrophoresis. Additionally, we assessed RNA purity based on OD260/280 and OD260/230 ratios using a NanoPhotometer spectrophotometer. RNA concentration was precisely quantified using a Qubit 2.0 fluorometer, and RNA integrity was accurately assessed using an Agilent 2100 Bioanalyzer.

Next, we employed fastp software version 0.22.0 [86] for quality control of raw reads. Reads containing adapters and low-quality data were filtered out, including the following: reads containing adapter sequences, reads with more than 10% undetermined bases (N), reads consisting entirely of A bases, and reads with over 50% of bases having quality scores (Q) ≤ 20. Subsequently, the remaining short reads were aligned to internal reference genes using the bowtie2 alignment tool [87], while HISAT2 software version 2.2.1 [88] was employed to map the reads to the reference genome. The data were plotted using the R (http://www.r-project.org/, accessed on 24 April 2024) ggplot2

The data processed by the PCA package version 4.4.0 beta were divided into 6 groups (CK-MF, CK-MS, MF 120 h, MF 24 h, MS 120 h, MS 24 h), and the data were visualized with the ggplot2 package version 3.4.2. The Corrplot package version 0.94 is used for Pearson correlation analysis and outputs heat maps.

### 4.4. Analysis of DEGs

After reconstructing complete transcripts using StringTie, genes identified in the sequencing results that were not present in the reference genome or reference gene set were classified as novel genes.

For differential expression analysis, we utilized DESeq2 version 1.30.1 [89] software. The analysis was conducted in three steps: read count normalization, calculation of hypothesis testing probability values (*p*-values) based on the model, and determination of false-discovery rate (FDR) values through multiple hypothesis testing. Significant DEGs were identified using the following criteria: FDR < 0.05 and |log2FC| > log2(2), where FC represents the fold change in expression. Statistical mapping of DEGs genes was performed using Excel, while Venny2.1 (https://bioinfogp.cnb.csic.es/tools/venny/index.html, accessed on 12 December 2023) was employed to assess the degree of coincidence among statistically significant DEGs.

### 4.5. GO and KEGG Enrichment Analyses

DEGs were annotated using the GO database (http://www.geneontology.org/, accessed on 12 December 2023) to determine the number of genes associated with each term and to perform GO function statistics. Subsequently, a hypergeometric test was employed to identify significantly enriched GO terms among the DEGs as compared to background. Visualization of the results included the generation of a histogram and a bubble map depicting GO enrichment classifications.

For pathway analysis, the KEGG database (https://www.genome.jp/kegg/, accessed on 12 December 2023) was utilized to assess pathway significance enrichment. Using KEGG pathways as the unit, a hypergeometric test was conducted to identify pathways significantly enriched among the DEGs as compared to the entire background. The results were visualized through a KEGG enrichment bar plot.

### 4.6. Validation of RNA-seq Data by Real-Time qRT-PCR (Quantitative RT-PCR)

DEGs identified using methods described above were validated using qRT-PCR. We screened genes expressed in both MS and MF, and randomly selected 6 genes with obvious up-regulation or down-regulation of expression after 24 h and 120 h of cold stress for RT-qPCR (*MsG0180005449*, *MsG0680035598*, *MsG0180001271*, *MsG0580027615*, *MsG0880046364* and *MsG0480023735*). *Medicago sativa* L. UBC2 and *Medicago sativa* L. TUB (GenBank accession no.L06967.1 and XM003630465) were used as internal reference genes to investigate the gene-expression levels of MS and MF after 24 h and 120 h of cold stress. Primer sets were designed using Primer Premier v6.24 (https://www.premierbiosoft.com/primerdesign/ accessed on 18 December 2023) and are listed in Additional file: Appendix A. Reactions were performed using the following conditions: denaturation at 95 °C for 30 s followed by 40 cycles of amplification (95 °C for 5 s, 60 °C for 34 s). All samples were tested in triplicate.

## 5. Conclusions

Both MS and MF displayed morphological changes in response to under 24-h and 120-h cold stress exposure. However MF exhibited superior resilience after 120 h, as evidenced by consistently lower levels of MDA and higher levels of SOD and SS. Also of note, MF demonstrated a greater increase in MF SS compared to MS after 24 h of cold stress.

Transcriptome analysis revealed 1274 and 2983 genes consistently regulated under 120 h of cold stress in MS and MF, respectively, with 923 genes shared between the two species. GO enrichment and KEGG analyses unveiled significant differences in genes responsive to cold stress between MS and MF. Particularly noteworthy was the enrichment of cell wall-related GO terms in MF, including “pectinesterase activity”, “cell wall modification”, and “cell wall”, emphasizing the role of the cell wall in providing stability and protection during cold stress.

Furthermore, while MS exhibited one unique pathway and one TF family member compared to MF, MF showcased 14 pathways and 9 TF family members absent in MS. Regardless, the expression of AP2-EREBP, ARR-B, and bHLH TFs was notable in both species, suggesting their potential role in conferring sustained cold resistance across other alfalfa species.

In summary, our comparative analysis of MS and MF cold stress responses provides valuable insights into molecular mechanisms underlying cold resistance in alfalfa. These findings significantly contribute to our understanding of cold stress tolerance in alfalfa and have far-reaching implications for breeding and genetic improvement efforts in related species within the clover genus.

## Figures and Tables

**Figure 1 ijms-25-10345-f001:**
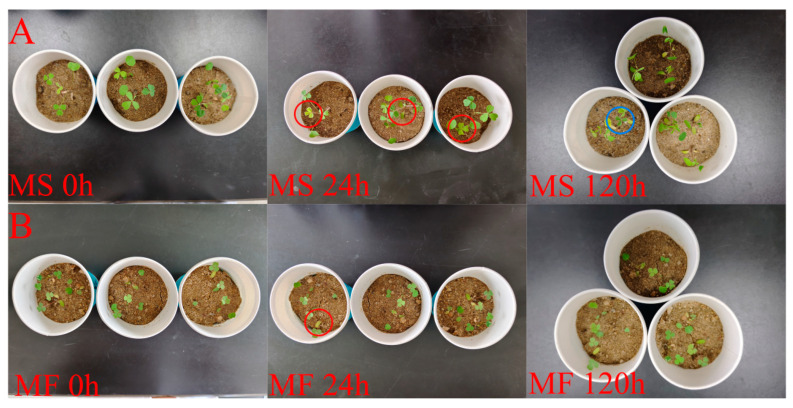
Morphological changes of MS and MF under cold stress. (**A**) MS was treated at 4 °C for 0 h, 24 h, and 120 h, respectively, and the leaves in the red circle were unhealthy while the leaves in the blue circle were healthy. (**B**) MF was treated at 4 °C for 0 h, 24 h, and 120 h, respectively, and the leaves in the red circle were unhealthy.

**Figure 2 ijms-25-10345-f002:**
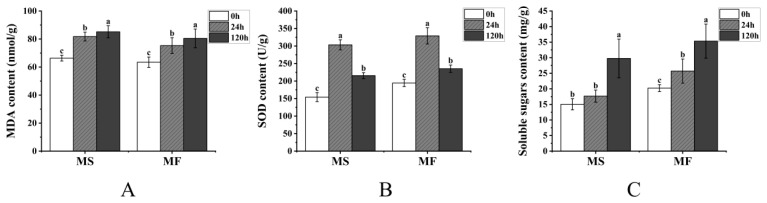
Contents of malondialdehyde (MDA), superoxide dismutase (SOD), and soluble sugar in the leaves of *Medicago sativa* L. and *Medicago falcata* L. under cold stress. (**A**) MDA content, (**B**) SOD content, and (**C**) soluble sugar content. Each value represents the mean of three replicates ± SD (standard deviation), represented by a vertical error bar. According to Duncan’s multiple range test, different letters above the bar indicate significant differences at the 0.05 level. Among them, a, b and c are the results of single-factor ANOVA analysis, which can reflect the significant degree of differences between treatments.

**Figure 3 ijms-25-10345-f003:**
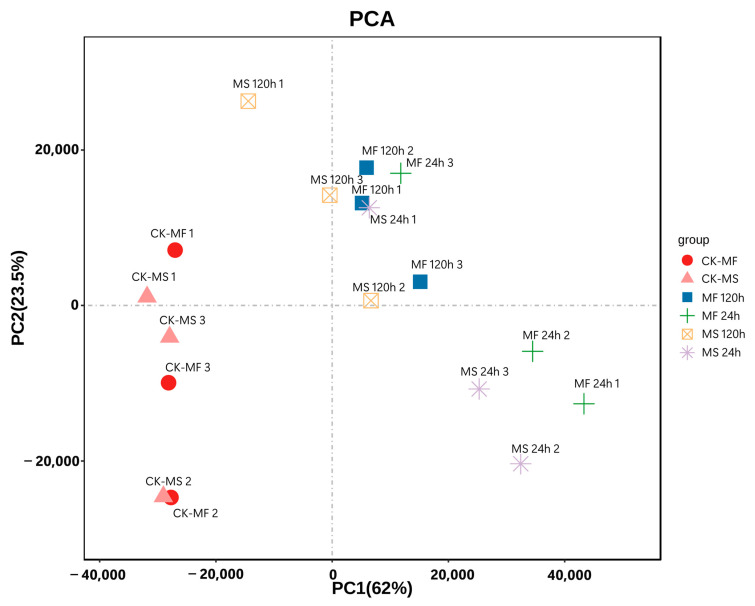
Principal component analysis (PCA) diagram of the sample. The PC1 coordinate represents the first principal component, and the percentage in parentheses indicates the contribution of the first principal component to the sample difference. The PC2 coordinates represent the second principal component, and the percentages in parentheses indicate the contribution of the second principal component to the sample variance. The colored dots in the diagram represent each sample.

**Figure 4 ijms-25-10345-f004:**
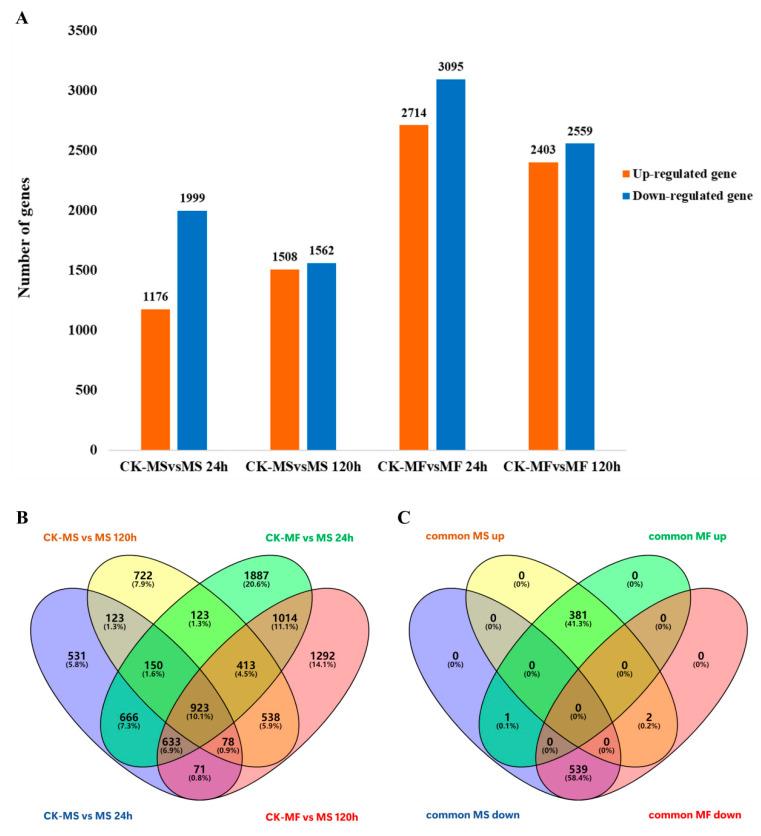
Analysis of DEGs. (**A**): Number of DEGs in each comparison group; (**B**): overlap of DEGs between two varieties compared with controls; and (**C**): overlap of two varieties co-expressed DEGs. Common MS up/down: up-regulated/down-regulated DEGs in *Medicago sativa* L.; common MF up/down: DEGs up-regulated/down-regulated in *Medicago falcata* L.

**Figure 5 ijms-25-10345-f005:**
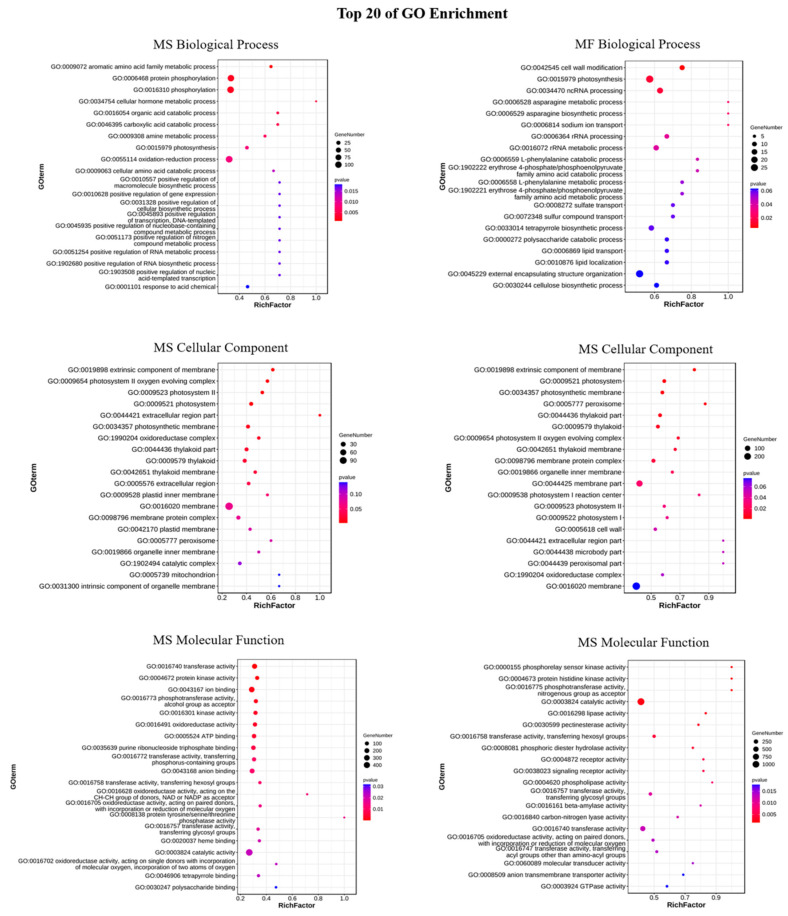
GO enrichment analysis of DEGs in *Medicago sativa* L. and *Medicago falcata* L. The GO enrichment bubble diagram of DEGs in MS and MF were plotted using the top 20 GO terms with the smallest Q values. The vertical coordinate was the GO term, and the horizontal coordinate was the enrichment factor (the number of differential genes in the GO term divided by all the numbers in the GO term). The size represented the number of genes, and the redder the color, the smaller the Q value).

**Figure 6 ijms-25-10345-f006:**
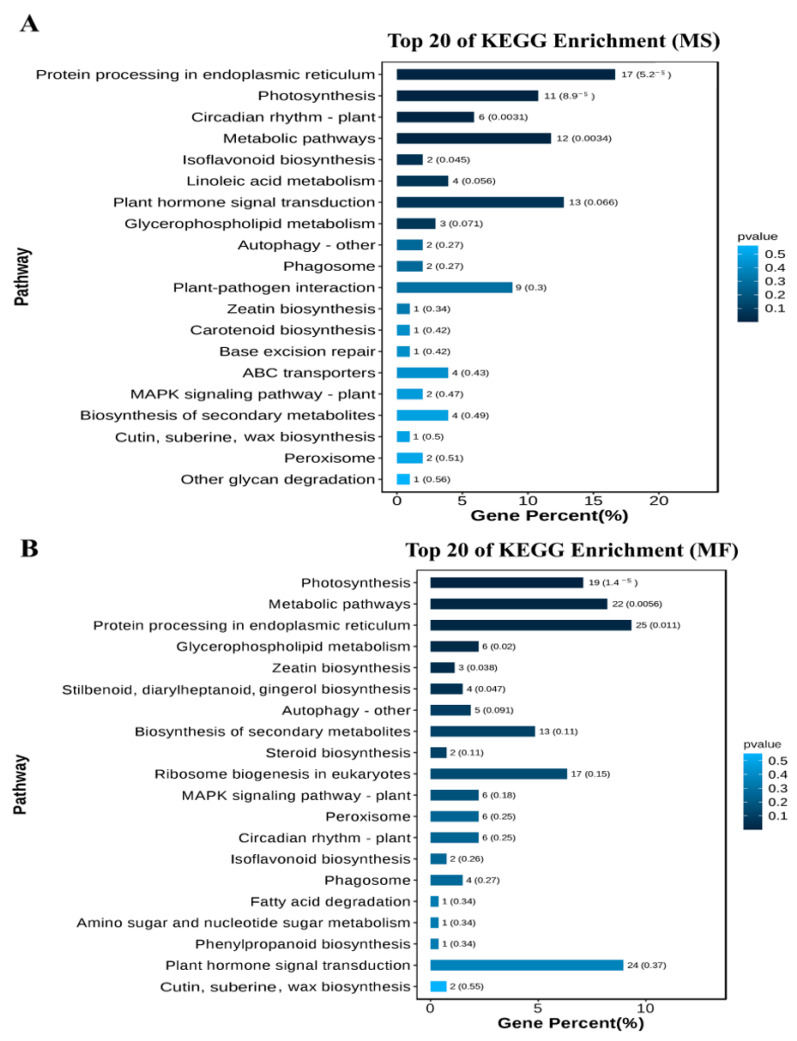
KEGG enrichment analysis of DEGs in *Medicago sativa* L. and *Medicago falcata* L. (**A**): The KEGG pathway enrichment of DEGs in *Medicago sativa* L. was mapped using the top 20 pathways with the smallest Q values. The ordinate is the pathway and the horizontal coordinate is the percentage of the number of this pathway in the number of all the different genes. The darker the color, the smaller the Q value, and the values on the column are the number and Q value of the pathway. (**B**): KEGG pathway enrichment of DEGs in *Medicago falcata* L.

**Figure 7 ijms-25-10345-f007:**
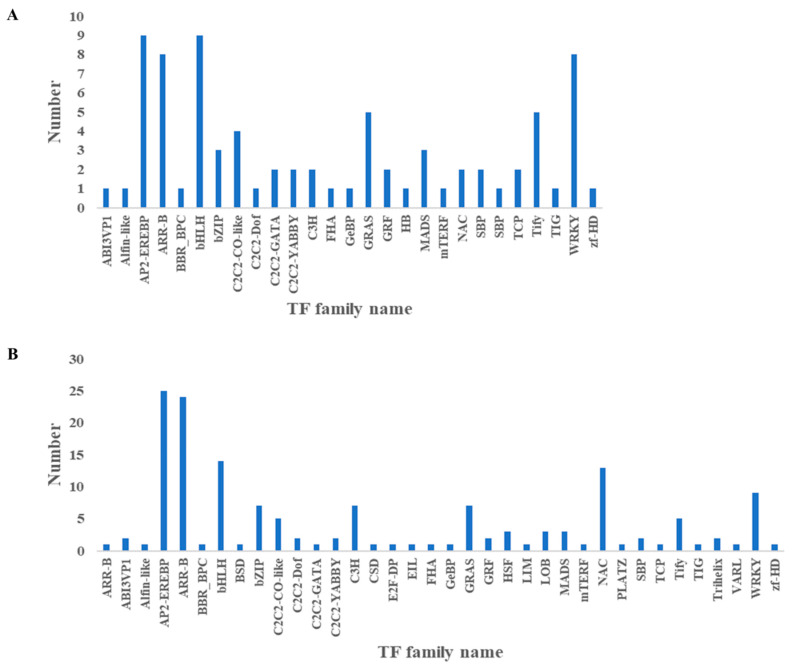
Family distribution of transcription factors in cold-stressed DEGs of *Medicago sativa* L. and *Medicago falcata* L. (**A**): Transcription factors in *Medicago sativa* L. (**B**): Transcription factors in *Medicago falcata* L.

**Figure 8 ijms-25-10345-f008:**
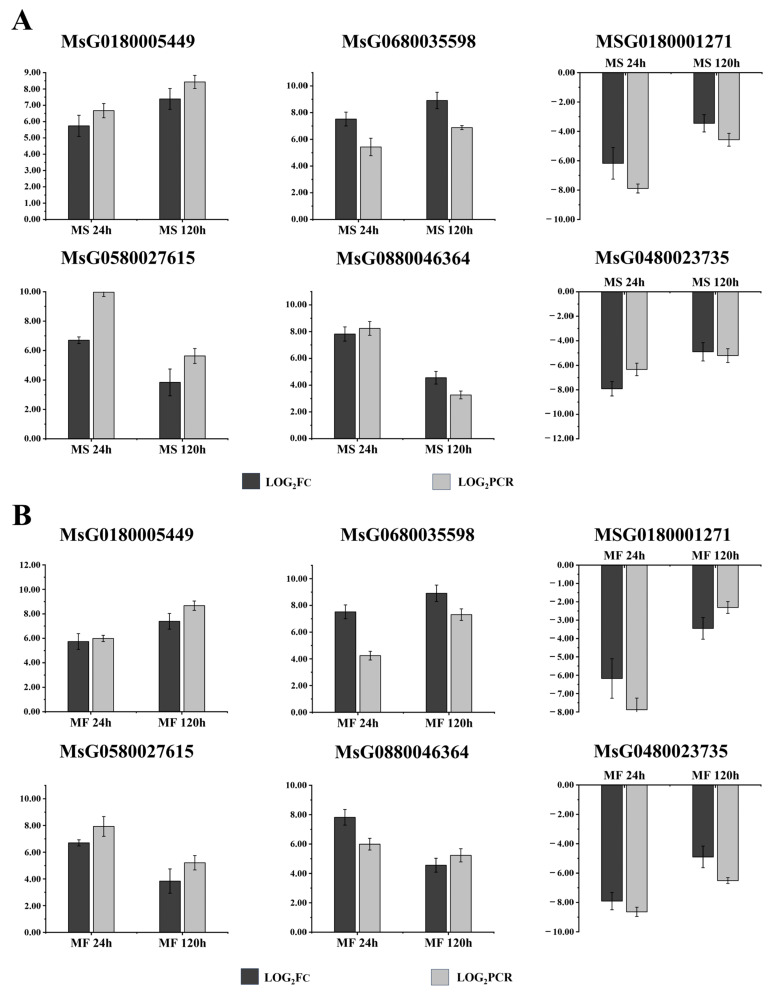
Validation of the RNA-seq data expression profile by qRT-PCR. The relative expression levels of six DEUs were calculated according to the 2^−ΔΔCt^ method using the *Medicago sativa* L. UBC2 and *Medicago sativa* L. TUB (GenBank accession no. L06967.1 and XM003630465) gene as an internal reference gene. We converted the obtained 2^−ΔΔCt^ values with the log formula to obtain what LOG_2_FC and LOG_2_PCR represented. Error bars represent standard deviations. (**A**) and (**B**) show the changes in genes in MS and MF, respectively. The x-axis indicates the various exposure durations of 4 °C cold stress: 24 h (MS/MF 24 h) and 120 h (MS/MF 120 h).

**Table 1 ijms-25-10345-t001:** Statistics and comparison of sequencing data of each sample.

Sample	Raw Data (Gb)	Clean Data (Gb)	Clean Reads	Q30 (%)	Q20 (%)	GC (%)	Unique_Mapped (%)	Multiple_Mapped (%)	Total_Mapped (%)
CK-MS 1	5.62	5.59	36,485,352	94.05	98.10	42.44	74.64	3.71	78.35
CK-MS 2	6.35	6.31	40,096,186	93.99	98.04	42.95	74.32	3.58	77.90
CK-MS 3	5.47	5.44	32,595,076	94.35	98.19	43.21	74.73	3.80	78.53
CK-MF 1	6.31	6.27	40,325,286	93.88	97.99	42.09	72.73	3.71	76.44
CK-MF 2	6.27	6.22	39,122,956	94.62	98.29	43.11	72.38	3.61	75.99
CK-MF 3	6.28	6.23	37,066,462	94.21	98.10	43.32	71.57	4.06	75.63
MS 24 h 1	6.31	6.26	38,466,796	93.09	97.65	42.82	75.25	4.41	79.66
MS 24 h 2	6.24	6.20	41,194,666	93.89	97.99	42.21	76.20	3.72	79.92
MS 24 h 3	6.07	6.02	37,839,116	93.86	97.98	42.77	75.67	4.08	79.75
MF 24 h 1	6.22	6.18	40,993,790	93.80	97.93	42.25	72.91	3.86	76.76
MF 24 h 2	5.56	5.53	33,894,138	93.87	97.98	42.98	72.15	4.17	76.32
MF 24 h 3	6.16	6.13	39,815,090	93.88	97.99	41.94	72.90	3.98	76.88
MS 120 h 1	6.21	6.16	38,873,226	94.20	98.11	42.60	74.97	4.41	79.38
MS 120 h 2	6.27	6.22	38,736,768	94.09	98.05	43.00	74.53	4.11	78.64
MS 120 h 3	6.55	6.50	39,479,598	93.45	97.78	43.05	74.44	4.36	78.80
MF 120 h 1	5.50	5.46	34,913,352	93.94	98.04	42.39	72.12	4.04	76.15
MF 120 h 2	5.64	5.61	34,576,766	93.81	97.96	42.73	71.20	4.32	75.53
MF 120 h 3	6.10	6.06	39,593,770	94.30	98.18	42.05	72.24	4.06	76.30
Average	6.06	6.02	38,003,800	93.96	98.02	42.66	73.61	4.00	77.61

Note: CK-MS, MS 24 h, and MS 120 h represent 0 h, 24 h, and 120 h cold-stress treatment groups of *Medicago sativa* L., respectively. CK-MF, MF 24 h, and MF 120 h represent 0 h, 24 h, and 120 h cold stress treatment groups of *Medicago falcata* L., respectively. DEGs under cold stress.

**Table 2 ijms-25-10345-t002:** Statistics on gene alignment.

Sample	Refer_Genes	Sequenced_Refer_Genes (%)	Novel_Genes	Sequenced_Novel_Genes (%)	Total_Genes	Sequenced_Total_Genes (%)
CK-MS 1	49,165	21,918 (44.58%)	7915	5171 (65.33%)	57,080	27,089 (47.46%)
CK-MS 2	49,165	21,401 (43.53%)	7915	4989 (63.03%)	57,080	26,390 (46.23%)
CK-MS 3	49,165	21,364 (43.45%)	7915	4963 (62.70%)	57,080	26,327 (46.12%)
CK-MF 1	49,165	21,775 (44.29%)	7915	5312 (67.11%)	57,080	27,087 (47.45%)
CK-MF 2	49,165	21,203 (43.13%)	7915	5003 (63.21%)	57,080	26,206 (45.91%)
CK-MF 3	49,165	21,410 (43.55%)	7915	4941 (62.43%)	57,080	26,351 (46.17%)
MS 24 h 1	49,165	22,001 (44.75%)	7915	5121 (64.70%)	57,080	27,122 (47.52%)
MS 24 h 2	49,165	21,823 (44.39%)	7915	5283 (66.75%)	57,080	27,106 (47.49%)
MS 24 h 3	49,165	22,141 (45.03%)	7915	5401 (68.24%)	57,080	27,542 (48.25%)
MF 24 h 1	49,165	21,113 (42.94%)	7915	5071 (64.07%)	57,080	26,184 (45.87%)
MF 24 h 2	49,165	21,271 (43.26%)	7915	5360 (67.72%)	57,080	26,631 (46.66%)
MF 24 h 3	49,165	21,473 (43.68%)	7915	5301 (66.97%)	57,080	26,774 (46.91%)
MS 120 h 1	49,165	21,296 (43.32%)	7915	4652 (58.77%)	57,080	25,948 (45.46%)
MS 120 h 2	49,165	21,258 (43.24%)	7915	4881 (61.67%)	57,080	26,139 (45.79%)
MS 120 h 3	49,165	21,520 (43.77%)	7915	5019 (63.41%)	57,080	26,539 (46.49%)
MF 120 h 1	49,165	20,948 (42.61%)	7915	5265 (66.52%)	57,080	26,213 (45.92%)
MF 120 h 2	49,165	21,137 (42.99%)	7915	5227 (66.04%)	57,080	26,364 (46.19%)
MF 120 h 3	49,165	21,524 (43.78%)	7915	5208 (65.80%)	57,080	26,732 (46.83%)
All	49,165	31,837 (64.76%)	7915	7915 (100.00%)	57,080	39,752 (69.64%)

Note: The numbers in the last row represent the sum of all the different genes.

## Data Availability

Data is contained within the article and Appendix A.

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
