# Peer review of "Identification of Cold Tolerance Transcriptional Regulatory Genes in Seedlings of *Medicago sativa* L. and *Medicago falcata* L."

_ijms, 2024, doi:10.3390/ijms251910345_

Round 1

Reviewer 1 Report

Comments and Suggestions for Authors

In this manuscript, the authors compared the transcriptome profiling between Medicago sativa L. and Medicago falcata L. The results of this comparison can provide insights into genes that could potentially enhance the cold tolerance of alfalfa through genetic improvement. In my opinion, there is still room for improvement in revising this paper.

1. The authors should provide phenotypic photos of the MS and MF after they have been exposed to cold stress.

2. Based on the physiological responses presented in Figure 1, it is difficult to substantiate the claim in the abstract that 'Our investigation unveiled superior physiological cold stress resilience in MF'.

3. Based on the results from table 1. The total mapping rate from all the samples were below 80%. Why?

4. Based on the PCA result, the reproducibility between biological replicates is poor. The authors should explain how they plan to improve the data in follow-up experiments.

5.  For part 2.4, the authors should provide a more comprehensive explanation of the GO enrichment results, going beyond merely focusing on the number of enriched GO items.

6. For Figure 7, the authors should enhance its resolution to ensure clarity and readability. The current version is not sufficiently detailed and makes it difficult to interpret the information presented.

7. Line 235, the authors should explain why the six candidate genes were selected.

8. Line 569-572, the authors should carefully check “The PCA()”, “ggplot2()”, “Corrplot()”.

Author Response

Dear Reviewer,

Thank you very much for your comments and professional advice. We have made corrections and revisions to the manuscript, and we hope that our work has improved this article. Furthermore, the details as follows:

Q1. The authors should provide phenotypic photos of the MS and MF after they have been exposed to cold stress.

Answer: We have supplemented the photos of the MS and MF, as shown in Figure 1, and marked the obvious differences listed in lines 116 to 120.

Q2. Based on the physiological responses presented in Figure 1, it is difficult to substantiate the claim in the abstract that 'Our investigation unveiled superior physiological cold stress resilience in MF.

Answer: We have added the photographs of the MS and MF phases as Figure 1. And we've revised the abstract to make it more rigorous while supplementing the plant morphology map, which were listed in lines 21 to 28 marked with yellow. (Our study revealed that MF had superior physiological resilience to cold stress compared with MS, and its morphology was healthier under cold stress, and its malondialdehyde content and superoxide dismutase activity increased first and then decreased, while the soluble sugar content continued to accumulate. Transcriptome analysis showed that after 120 hours of exposure, there were different gene expression patterns between MS and MF, including 1274 and 2983 genes that were continuously upregulated, respectively, and a total of 923 genes were included, including star cold-resistant genes such as ICE1 and SIP1.)

Q3. Based on the results from table 1. The total mapping rate from all the samples were below 80%. Why?

Answer: Thank you very much for asking a valuable question, our plant material was cultivated from 45 degrees north latitude, which makes them somewhat different from the reference genome. We explained this in the text of lines 301 to 304 marked with yellow. (The overall comparison rates between the materials selected in this study and the alfalfa reference genome ranged from 75.53% to 79.92%, likely due to the cold region specificity of the plant material we used being produced at 45 degrees north latitude.)

Q4. Based on the PCA result, the reproducibility between biological replicates is poor. The authors should explain how they plan to improve the data in follow-up experiments.

Answer: Based on the results of PCA analysis, we have made some changes and additions to the text, which were listed in yellow in lines 152 to 155. (The results of principal component analysis (PCA) showed that the biological replicates were almost identical between the treatment groups, and in subsequent analyses we will focus on individuals with good reproducibility as references.)

Q5. For part 2.4, the authors should provide a more comprehensive explanation of the GO enrichment results, going beyond merely focusing on the number of enriched GO items.

Answer: We've taken this great idea and added important information about the GO and KEGG analysis in lines 200 to 204 and 209 to 211. (The most abundant GO terms for MS and MF in biological processes were "oxidation-reduction process (GO:0055114)" and "external encapsulating structure organization (GO:0045229)", respectively. And the most numerous GO items in their cellular components and molecular functions were "membran (GO:0016020)" and "catalytic activity (GO:0003824)". The three pathways with the highest number of genes in MS and MF are "Metabolic pathways", "Protein processing in endoplasmic reticulum", "Plant hormone signal transduction".)

Q6. For Figure 7, the authors should enhance its resolution to ensure clarity and readability. The current version is not sufficiently detailed and makes it difficult to interpret the information presented.

Answer: Thank you very much for your feedback, this is really an issue to look out for. We remastered Figure 7 and this time using the TIFF format, which made it much more resolution.

Q7. Line 235, the authors should explain why the six candidate genes were selected.

Answer: Thank you for your question, and we have added the reasons for the selection of these six genes to the Materials & Methods, which were listed in lines 592 to 598 marked with yellow. (We screened genes expressed in both MS and MF, and randomly selected 6 genes with obvious up-regulation or down-regulation of expression after 24 h and 120 h of cold stress for RT-qPCR (MsG0180005449, MsG0680035598, MsG0180001271, MsG0580027615, MsG0880046364 and MsG0480023735). Medicago sativa L. UBC2 and Medicago sativa L. TUB (GenBank accession no.L06967.1 and XM003630465) were used as internal reference genes to investigate the gene expression levels of MS and MF after 24 h and 120 h of cold stress.)

Q8. Line 569-572, the authors should carefully check “The PCA()”, “ggplot2()”, “Corrplot()

Answer: Thanks for your question, we've revised this paragraph to make it clearer, and listed in lines 561 to 564 with yellow. (The data processed by the PCA package were divided into 6 groups (CK-MF, CK-MS, MF 120h, MF 24h, MS 120h, MS 24h), and the data were visualized with the ggplot2 package. The Corrplot package is used for Pearson correlation analysis and outputs heat maps.)

In addition, we have fleshed out the discussion by analyzing the important role of the ICE1-CBF-COR pathway and aquaporins in the process of resistance to cold stress, so that this article is not limited to the comparison of MS and MF.

Thank you very much for your attention and time. Look forward to hearing from you.

Yours sincerely,

16-Aug., 2024

Reviewer 2 Report

Comments and Suggestions for Authors

The paper is interesting an describes a complete study on the genes related to cold response in two medicago species. The paper has interest to the readers and provides a lot of interesting novel informations, but some aspects need to be improved prior to acceptance.

The main problem I found has to be with the discussion. Authors center the discussion on the differences between the two species... but how this novel data relates with our knowledge on cold response in other plants.

For instance:

The ice and eskimo genes.. are also regulated in medicago?

Aquaporins have been shown to be important in cold response... how many DEGS represent aquaporins? Which ones? Are conserved in other plants? Please include this in the discussion. 

Figures 1, 4 and 7 need to improve the quality, as nothing can be read. Plase enlarge th font size.

Author Response

Dear Reviewer,

Thank you very much for your comments and professional advice. We have made corrections and revisions to the manuscript. Furthermore, the details were shown as follows:

Q1. The main problem I found has to be with the discussion. Authors center the discussion on the differences between the two species... but how this novel data relates with our knowledge on cold response in other plants.

For instance:

The ice and eskimo genes.. are also regulated in medicago?

Aquaporins have been shown to be important in cold response... how many DEGS represent aquaporins? Which ones? Are conserved in other plants? Please include this in the discussion. 

Answer: Thank you very much for your comments, which are very valuable in elevating our articles. We screened and identified DEGs of the ICE genes and the CBF genes between MS and MF, and the ICE1-CBF-COR pathway is a very important cold-resistant pathway, and it is also the most widely studied. We found the ICE1 gene in MS and MF, and the CBF5 gene in MF, and they were both up-regulated, which is consistent with the results of previous studies on other crops, indicating that both the ICE genes and the CBF genes regulate the process of alfalfa cold resistance.

In addition, we also found a total of 4 isoforms of aquaporins in MS and MF, among which the SIP genes are only present in MF, which is very valuable for our research. Among the four isoforms, the genes regulating TIP were continuously down-regulated, and the genes regulating SIP were continuously up-regulated, which was consistent with the results of previous studies, indicating that these two genes were activated by low-temperature conditions and played an important role in cold domestication. In the past, the PIP genes were verified to be up-regulated with cold stress, but in our experiment, the PIP genes in MS and MF was down-regulated, and the function of PIP genes is related to the plasma membrane, so we suspect that the plasma membrane is damaged due to prolonged cold stress, and this part needs to be further studied.And we also found an interesting phenomenon: the NIP genes were down-regulated in MS, down-regulated and up-regulated in MF, and the NIP genes down-regulated in MF gradually recover a little in the four hours between 24 h and 120 h of cold stress, but still did not reach the up-regulation overall. This phenomenon has also been observed in past studies, for example, with the progress of cold stress, the NIP genes in chickpea and grape are up-regulated, but the NIP genes in tea plant are down-regulated. So far, there have been no reports of both up-regulation and down-regulation of the NIP genes in the same plant species, so we think our findings are very interesting, but further research is needed.

We have supplemented our findings in lines 390 to 434 and marked with yellow.

Q2. Figures 1, 4 and 7 need to improve the quality, as nothing can be read. Plase enlarge th font size.

Answer: Thank you very much for your feedback, this is really an issue to look out for. We remastered Figure 7 with the TIFF format, which made it much more resolution.

We've also corrected and added some of the less rigorous content, which were marked with yellow areas of the text.

Thank you very much. Look forward to hearing from you.

Yours sincerely,

16-Aug.,2024

Round 2

Reviewer 1 Report

Comments and Suggestions for Authors

In my opinion, this manuscript still has room for improvement.

1.       For Figure 8, the authors should explain what LOG2FC and LOG2QTL represent. In addition, it seems that these results were not based on the 2−ΔΔCt method. The authors should carefully review this.

2.       The authors should carefully verify that the description "Table 2. Genetic testing statistics." is accurate.

3.       According to Figure 3, the reproducibility among biological replicates appears to be suboptimal. The authors should provide an explanation of the measures they intend to take in subsequent experiments to enhance the data quality. For instance, the "CK-MF1", "CK-MF2", and "CK-MF3" groups are not tightly clustered, and the "MS 24H 1" group is distant from "MS 24H 2" and "MS 24H 3", indicating a lack of consistency. The authors should address these issues and outline their strategy for improving reproducibility in future experiments.

Author Response

Dear Reviewer,

Thank you very much for your comments and professional advice. We have completed the revision of the manuscript according to your suggestions. Here list the details as follows:

Q1. For Figure 8, the authors should explain what LOG2FC and LOG2QTL represent. In addition, it seems that these results were not based on the 2−ΔΔCt method. The authors should carefully review this.

Answer: Thank you very much for spotting where we are lacking. We recalculated the data using 2−ΔΔCt method, then used the log formula to transform the results and create a new image. LOG2FC represents the 2−ΔΔCt value of the gene of interest after the log formula, and LOG2QTL has been changed to LOG2PCR to represent the 2−ΔΔCt value of the housekeeping genes of interest after the log formula. We've added explanations of LOG2FC and LOG2PCR to the legend, and the new images and additions were listed in lines 246 to 251 of the text. In addition to this, we have added information such as temperature and number of cycles used in PCR. (We converted the obtained 2−ΔΔCt values with the log formula to get what LOG2FC and LOG2PCR represented. )

Q2. The authors should carefully verify that the description "Table 2. Genetic testing statistics." is accurate.

Answer: We believe that your queries are very valuable, so we have provided a more precise description of Table 2 and added a note to it which were listed in green on lines 174 to 175 of the text.

Q3. According to Figure 3, the reproducibility among biological replicates appears to be suboptimal. The authors should provide an explanation of the measures they intend to take in subsequent experiments to enhance the data quality. For instance, the "CK-MF1", "CK-MF2", and "CK-MF3" groups are not tightly clustered, and the "MS 24H 1" group is distant from "MS 24H 2" and "MS 24H 3", indicating a lack of consistency. The authors should address these issues and outline their strategy for improving reproducibility in future experiments.

Answer: Thank you for your valuable comments. As we mentioned in lines 143 to 145, in this study, we used the group with better reproducibility of PCA results as the primary analysis object and the other group as the reference. At the same time, we have added plans for the future, and in future experiments, we will further refine the mechanism of alfalfa resistance to cold at the transcriptional level and increase biological and technical duplication to improve data quality. The places where the changes made were shown in blue on line 292 to 296 of the text. (In this study, we used the group with better reproducibility of PCA results as the main analysis object, and the other group as a reference. Next, we will further refine the mechanism of alfalfa resistance to cold at the transcriptional level, and increase biological and technical replicates to improve the quality of the data.)

Thank you again.

Yours sincerely,

18-Aug., 2024

Round 3

Reviewer 1 Report

Comments and Suggestions for Authors

As mentioned in the previous comments, the authors stated, ' In this study, we used the group with better reproducibility of PCA results as the main analysis object, and the other group as a reference. Next, we will further refine the mechanism of alfalfa resistance to cold at the transcriptional level, and increase biological and technical replicates to improve the quality of the data.' However, according to Figure 3, the three biological replicates from 'CK-MF' and 'MS 120h' exhibit significant variation among themselves, raising the question of which replicate was chosen for subsequent analysis.

Author Response

Dear Reviewer,

Thank you very much for your comments and professional advice. We have completed the revision of the manuscript according to your suggestions. Here list the details as follows:

Q1. As mentioned in the previous comments, the authors stated, ' In this study, we used the group with better reproducibility of PCA results as the main analysis object, and the other group as a reference. Next, we will further refine the mechanism of alfalfa resistance to cold at the transcriptional level, and increase biological and technical replicates to improve the quality of the data.' However, according to Figure 3, the three biological replicates from 'CK-MF' and 'MS 120h' exhibit significant variation among themselves, raising the question of which replicate was chosen for subsequent analysis..

Answer: We followed the principles: if there was no significant difference between the three replicates, the average was taken for subsequent analysis; If there was a deviation in one replicate, the two most concentrated replicates were selected and the average was taken for calculation.

Thank you again.

Yours sincerely,

19-Aug., 2024
